# Suicide Prevention Program with Cooperation from Senior Volunteers, Governments, and Schools: A Study of the Intervention Effects of “Educational Lessons Regarding SOS Output” Focusing on Junior High School Students

**DOI:** 10.3390/children9040541

**Published:** 2022-04-11

**Authors:** Susumu Ogawa, Hiroyuki Suzuki, Tomoya Takahashi, Koji Fujita, Yoh Murayama, Kenichiro Sato, Hiroko Matsunaga, Yutaka Motohashi, Yoshinori Fujiwara

**Affiliations:** 1Research Team for Social Participation and Community Health, Tokyo Metropolitan Institute of Gerontology, 35-2 Sakae-cho, Itabashi-ku, Tokyo 173-0015, Japan; s_ogawa@tmig.or.jp (S.O.); suzukihy@tmig.or.jp (H.S.); ttkhs@tmig.or.jp (T.T.); fujita@tmig.or.jp (K.F.); yhoyho05@tmig.or.jp (Y.M.); sato_ken@tmig.or.jp (K.S.); hiroko_m@tmig.or.jp (H.M.); 2National Center of Neurology and Psychiatry, 4-1-1 Ogawa-Higashi, Kodaira, Tokyo 187-8502, Japan; motohasiy@ncnp.go.jp

**Keywords:** SOS program, senior volunteer, picture-book reading, local government, junior high school

## Abstract

As a suicide countermeasure for young people, implementing “SOS output education” that provides young people with opportunities and approaches to seeking support with community cooperation can be expected to reduce lifelong suicide risk. We implemented an “SOS output education” for junior high school students with cooperation from educators, government staff, and older people working as community volunteers. A total of 188 students were allocated to an intervention group and a waiting group. Outcome assessments were implemented at three points in time: before the program (Time 1), after the program (Time 2), and three months after the program (Time 3). Results showed that the number of people with worries increased in the intervention group compared with the waiting group between Time 1 and Time 2. There was also an increase in people with “reliable adults” between Time 1 and Time 3, and people with “adults who you can talk to at any time” increased between Time 2 and Time 3 in the intervention group. By implementing the SOS output education program with community cooperation, an increase was observed in the intervention group in terms of support-seeking awareness and the number of people with reliable adults and with adults who they can talk to at any time.

## 1. Introduction

The 2019 Japanese suicide rate was 15.7 (per 100,000), which is the highest rate in any developed nation (G7: Group of Seven) [1]. The suicide attempt rate among females was 14.0% and among males was 30.6%. Since the establishment of the Basic Law on Suicide Countermeasures in 2006, comprehensive suicide countermeasures have been promoted, and the number of deaths by suicide in Japan has greatly decreased since peaking in 2010. Yet, there has been no decrease in the number of deaths by suicide among children and young people; rather, there has been an upward trend, which means that measures to combat youth suicide are an urgent issue.

In the United States, intervention studies with the purpose of preventing suicide among young people are being conducted by the Suicide Prevention Resource Center (SPRC) [2]. Studies were conducted about Signs of Suicide (SOS) among students in middle school (ages 11–13) and high school (ages 13–17), and intervention programs that teach students to be aware of the signs and to take action weres reported to potentially be effective in reducing suicidal ideation and planning and in improving knowledge of mental health [3].

In the Saving and Empowering Young Lives in Europe (SEYLE) study conducted in various European countries [4], gatekeeper programs were implemented among teachers, health education programs were implemented with students, and screening programs were implemented with specialists (comprising 5 h across 4 weeks). Among these, the program that showed the greatest effects was the psychological education program for students, which reduced suicidal planning and suicidal ideation 12 months thereafter.

In the SEYLE study [4], although it was confirmed that the intervention effects included raised levels of awareness and knowledge among participants that they could seek support when troubled, the study did not verify whether the intervention was effective in enabling participants to recognize where they could seek support. An important point in preventing suicide among school-aged youth is having prior awareness of public organizations from which they can seek support when they are struggling. However, even if the person is aware of public organizations from which they can seek support, some school-aged youth will hesitate to reach out because of perceived barriers. In such cases, the presence of trusting relationships with familiar, reliable adults, such as family members and others in the community, are important.

In Japan, too, there is a demand for the development and implementation of suicide prevention programs for school-aged young people [5]. The Ministry of Health, Labor and Welfare stated Comprehensive Measures to Prevent Suicide that serve as the guidelines for measures to prevent suicide that are promoted by the government [6]. These guidelines are different from the Signs of Suicide (SOS) in that the aim is not to decrease suicidal ideation and planning but to increase the number of reliable adults (i.e., people that youth can turn to) in the community who can respond when a young person is manifesting SOS [7].

In Japan, REPRINTS^®^ (research of productivity by intergenerational sympathy) [8] is a known initiative for the creation of intergenerational coexistence, mutual benefit, and social capital in communities. Modeled on research on Experience Corps^®^, a school volunteer intervention implemented by older people in slum areas in the United States [9,10], REPRINTS^®^ is a volunteer project developed by the Tokyo Metropolitan Institute of Gerontology in 2004 where older people read picture-books to young people. REPRINTS^®^ is an intervention program that promotes intergenerational exchanges between all sorts of groups within the community (children, students, parents, teachers, senior volunteers, etc.) and is not limited to older and young people [11]. Through picture-book reading, changes are seen in the impression formed and the attitude of elementary school students toward older people. Improved levels of awareness are also observed in junior high school students with regard to communicating with older people [12,13]. Changes in the impressions and attitudes formed among school-aged youth in relation to older people can be expected to inhibit ageism and contribute to the creation of social capital. In fact, a study by Murayama et al. (2019) found that there was an increase in regional social capital due to REPRINTS^®^ activities over the course of more than 10 years [14]. The concept of REPRINTS^®^ is consistent with the Japanese government’s Comprehensive Measures to Prevent Suicide. In Japan, too, there is a demand for the development and implementation of suicide prevention programs for school-aged young people.^5^ The Ministry of Health, Labor and Welfare stated intergenerational exchanges with older people in the community at school can promote SOS output education in its Comprehensive Measures to Prevent Suicide that serve as the guidelines for measures to prevent suicide that are promoted by the government [6], and so senior volunteers who participate in REPRINTS^®^ activities are playing a part in promoting measures to combat suicide.

This study aims to investigate the effects, in terms of measures to reduce student suicides and in terms of support-seeking activities among students, by clarifying the impact of picture-book-reading programs, working in collaboration with local government staff (government), educators (schools), and senior volunteers (over 65 years old) (communities).

## 2. Materials and Methods

### 2.1. Participants

This program was implemented in a lesson called “comprehensive learning time” with all second-year students (188 students in 5 classes) at Junior High School A in the Fuchu municipality of Tokyo.

### 2.2. Procedure

This study was implemented from June to September in 2019 (before the COVID-19 pandemic). The program was implemented on 25 September 2019 (intervention group) and on 16 July 2019 (waiting group).

Prior to program implementation, and with reference to “The Adachi Ward, Tokyo, model as a prototype” [5] that had already been conducted by the Tokyo Metropolitan Institute of Gerontology, a program was developed in which senior volunteers read picture-books to students. Regarding the picture-books read to students by senior volunteers, three volumes were selected as prospective picture-books to be read on the day from among the SOS Picture-Book List investigated and produced by the working group of REPRINTS^®^ members that was established in 2017. Later, with cooperation between the Tokyo Metropolitan Institute of Gerontology study team, staff from the Fuchu municipal health and welfare department, and school teachers, one of the three books was selected for use in sessions, namely, *You Are Special*, written by Max Lucado and illustrated by Sergio Martinez. After the program was established, local government staff and senior volunteers (REPRINTS^®^ members) cooperated in August 2018 to carry out rehearsals for educational lessons based on the program on the topic of SOS outputs. In June 2019, local government staff, school teachers, and senior volunteers cooperated to implement educational lessons on SOS outputs.

The scope covered 188 second-year students (14 years old) at Junior High School A in Fuchu, Tokyo. School teachers explained the lessons to students and their parents/guardians in advance and distributed and collected a pre-study questionnaire survey two weeks before program implementation (Time 1). Three of the five classes were assigned to the intervention group (*n* = 113) and two classes to the waiting group (*n* = 75). The intervention group participated in the program after Time 1. Three weeks after implementing the program, a second survey was conducted with the intervention group and waiting group (Time 2). The waiting group participated in the program after Time 2. Seven weeks after implementing the program, a third survey was conducted with the intervention group and waiting group (Time 3). The interval between Time 1 and Time 2 was three weeks, and the interval between Time 2 and Time 3 was seven weeks (i.e., the summer vacation), and, as such, the intervals were not evenly spaced. For that reason, comparisons of the intervention group and the waiting group were conducted at two points: Time 1 and Time 2. Furthermore, for the intervention group only, the longitudinal changes between Time 1, Time 2, and Time 3 were studied (Figure 1).

### 2.3. Program

The program was structured as 50 min lessons, conducted with cooperation from local government staff, school teachers, and senior volunteers (Table 1). On days on which the program was implemented, school teachers prepared the classroom and went around talking to students in order to facilitate the start of the lesson by local government staff.

In Part 1, local government staff explained what the lesson would be about, and, in Part 2, they taught students about stress, stress coping, and SOS outputs. The Guidelines for the Promotion of Education About SOS Outputs distributed by the Tokyo Municipal Board of Education was used in the lessons [15]. In the lessons regarding SOS outputs, first, the students were given opportunities to reflect on “living life to the fullest from the day they were born until now”, which had the objective of increasing students’ self-esteem. Next, the program explained the “SOS of the heart” when one has feelings of sadness or unease, and an explanation was given of stress, stress coping, and ways of seeking help. The key message about ways of seeking help was “try to talk to at least three people”, which is based on the idea that we need to take a proactive approach to find people who understand us.

Then, in Part 3, senior volunteers read a picture-book (*You Are Special*, written by Max Lucado, illustrated by Sergio Martinez) to the students. The picture-book tells the story of a person who goes looking for people he can rely on after he loses his self-confidence because everyone makes fun of him. The book’s message is that there will be some difficulties on the path to finding people that one can rely on and that it is important to have people who care about you regardless of what others say. The message of the picture-book was the same as the message of the lectures given by the local government staff. Being read the picture-book was intended to give students a sense of companionship with the senior volunteers, in addition to helping to summarize the program as a whole.

In Part 4, local government staff distributed leaflets to students about support organizations, letting students know that they can reach out to these organizations whenever they are struggling. Finally, the program was brought to an end with the students completing an evaluation sheet that asked about their impressions of the lesson.

### 2.4. Outcome Measures

Students were asked whether they had any “worries”, the “content of those worries (studying, future path, friendships, romance, bullying, family, local people, local situation, and other)”, and about “familiar adults (reliable adults, adults who you can talk to at any time, adults who care about you, adults who greet you)”.

To measure self-esteem, the general self-worth subscale of the Japanese Edition of the Harter’s Perceived Competence Scale for Children (shortened version) [16] was used. General self-worth is defined in almost the same way as self-esteem [17]. The scale comprises 10 statements, including “I can do most things better than others”, “I can state my opinion with confidence”, and “I don’t think I have many good points” with a 4-grade assessment being sought for each question (1 = agree, 2 = somewhat agree, 3 = somewhat disagree, and 4 = disagree). Reverse processing was performed on some items so that higher scores overall indicated greater general self-worth. The range of scoring was from 10 to 40 points (Cronbach’s α was 0.83).

### 2.5. Analysis

An analysis was conducted, covering those who responded at Time 1, Time 2, and Time 3. A total of 170 students (90.4%) gave complete responses to “worries” and “familiar adults (reliable adults, adults who you can talk to at any time, adults who care about you, adults who greet you)”, while 145 students (77.1%) gave complete responses regarding general self-worth.

Students were asked about the details of each of their worries (studying, future path, friendships, romance, bullying, family, local people, local situation, and other). If a student responded in the affirmative to one or more worry items, they were counted as “having worries”. If they did not have any of the worries, they were counted as “not having worries”. The analysis examined changes over time in the ratios of responses to “Presence of worries: Yes/No” and “Presence of familiar adults (reliable adults, adults who you can talk to at any time, adults who care about you, adults who greet you): Yes/No”. An analysis was also conducted based on categorizing “studying” and “future path” as “school-related” and “friendships” and “romance” as “human relationships”.

There were two dependent values for each time point (Yes, No), and the change between two time points occurred in four patterns (i.e., Time 1 = Yes & Time 2 = No, Time 1 = Yes & Time 2 = No, Time 1 = Yes & Time 2 = No, Time 1 = Yes & Time 2 = Yes). A comparison was made between the intervention group and waiting group in terms of the changes between Time 1 and Time 2. In order to find the interaction between groups and time points, statistical tests were applied using conditions that conform to approximate standard normal distribution on the basis of the null hypothesis that “the difference between the parameter ratios at Time 2 after intervention will be the same as between the two groups before the intervention”. In both tests, when |z > 1.96|, then *p* < 0.05.

The specific equations are shown below.
$$z=\frac{\left(\frac{a+c}{n_{group1}}-\frac{a+b}{n_{group1}}\right)-\left(\frac{a+c}{n_{group2}}-\frac{a+b}{n_{group2}}\right)}{\sqrt{\frac{n_{group1b}+n_{group1c}-\frac{{\left(n_{group1b}-n_{group1c}\right)}^{2}}{n_{group1}}}{n_{group1}^{2}}+\frac{n_{group2b}+n_{group2c}-\frac{{\left(n_{group2b}-n_{group2c}\right)}^{2}}{n_{group2}}}{n_{group2}^{2}}}}$$

Note: a: Time 1 = No & Time 2 = No, b: Time 1 = No & Time 2 = Yes, c: Time 1 = Yes & Time 2 = No, d: Time 1 = Yes & Time 2 = Yes.

Furthermore, for the intervention group, the changes between Time 1 and Time 3 were investigated using Cochran’s Q test.

In order to investigate interaction of the general self-worth, an analysis of variance (ANOVA) was performed for Group (intervention group, waiting group) × Times (Time 1, Time 2). A repeated measures ANOVA was performed on the changes between the three time points in the intervention group. The level of significance in this study was 5%, and Bonferroni correction was used for multiple comparisons. SPSS version 23 for Windows (IBM Inc., Chicago, IL, USA) was used for data analysis.

### 2.6. Ethical Approval

The ethics committee of the Tokyo Metropolitan Institute of Gerontology approved the authors to take and use the anonymized data from Fuchu city for secondary analysis (no. 25000, 27 December 2018).

## 3. Results

### 3.1. Group (Intervention, Waiting) × Time (Time 1, Time 2)

Table 2 shows the “presence of worries” at each of the three time points. First, regarding the overall presence of worries, a comparison was made of the changes in the ratios between Time1 and Time2 in each group. Interaction was found between groups and times with regard to the changes in the intervention group and waiting group (Z = 2.13, *p* < 0.05). There was an increase in the intervention group between Time 1 and Time 2 and a decrease in the waiting group. Regarding “school-related” and “human relationships”, no interaction was found between groups and times (Z = 0.78, n.s.; Z = 0.06, n.s.).

Table 3 shows the presence of familiar adults at the three time points. Regarding reliable adults, adults who you can talk to at any time, adults who care about them, and adults who greet them, no significant interaction was found in the changes of the ratios between groups or times (Z = 0.59, n.s.; Z = 0.27, n.s. Z = 1.83, n.s.; Z = 0.33, n.s.).

Regarding general self-worth (Table 4), the main effect of groups, the main effect of times, and interaction were not significant (F (1,154) = 0.01, ηp2 < 0.000; F (1,146) = 1.31, *p* = 0.255 n.s., ηp2 = 0.009; F (1,154) = 0.01, *p* = 0.937 n.s., ηp2 < 0.000).

### 3.2. Repeated Measures (Time 1, Time 2, Time 3)

For each of the variables in the intervention group, the changes between the three time points were investigated using Cochran’s Q test. The change in the ratio for “presence of worries” was significant (Q = 9.75, *p* = 0.008). The results of multiple comparison showed a decrease in those with worries between Time 2 and Time 3 (*p* = 0.007). Between Time 1 and Time 2 and between Time 1 and Time 3, there was no significant difference. There was significant difference in the change in the ratio of worries that were school-related (Q = 7.09, *p* = 0.029). The results of multiple comparison showed a decrease in the ratio of those with worries that were school-related between Time 2 and Time 3 (*p* = 0.032). Between Time 1 and Time 2 and between Time 1 and Time 3, there was no significant difference.

There was a significant difference in the change in the ratio of worries about human relationships (Q = 14.97, *p* < 0.001). The results of multiple comparison showed an increase in the ratio of those with worries about human relationships between Time 1 and Time 2 (*p* = 0.009) and a decrease between Time 2 and Time 3 (*p* = 0.001). There was no significant difference between Time 1 and Time 3.

The same analysis was conducted regarding the presence of familiar adults. The change in the ratio of “reliable adults” was significant (Q = 6.05, *p* = 0.048), and the significance increased between Time 2 and Time 3 (*p* = 0.047). There was no change between Time 1 and Time 2 or between Time 1 and Time 3. The change in the ratio of “adults who you can talk to at any time” was significant (Q = 6.35, *p* = 0.042), and the ratio was higher in Time 3 than in Time 1 (*p* = 0.035). There was no significant change between Time 1 and Time 2 or Time 2 and Time 3. The change in the ratio of “adults who care about you” and “adults who greet you” was not significant (Q = 0.65, *p* = 0.723; Q = 0.93, *p* = 0.629). For general self-worth, there was no main effect of time (F (2,182) = 0.15, *p* = 0.863, n.s., ηp2 = 0.002).

## 4. Discussion

The purpose of this study was to verify the effects of a training program to promote support-seeking behavior in students in which senior volunteers read picture-books to students with coordination between local government staff (government), junior high school teachers (schools), and senior volunteers (communities). Our results show that the program decreased the ratio of those with worries and increased the number of people with reliable adults and with adults who they can talk to at any time.

There was an increase in the ratio of those with worries about human relationships in the intervention group in comparison to those in the waiting group. In addition, with regard to the changes between the three time points in the intervention group only, while there was an increase between Time 1 and Time 2, there was a decrease between Time 2 and Time 3. It is possible that there was an increase in worries in the responses given because the students felt more able to disclose their worries after program implementation. Meanwhile, it is possible that a decrease was observed between Time 2 and Time 3 as a result of students being able to talk about their worries. The content of the picture-book used in this program was intended to raise students’ awareness of their ability to self-disclose their worries about human relationships, and so this may have been shown as an intervention effect.

No interaction was found regarding the change in the ratios in connection with worries that were school-related between the intervention group and the waiting group. However, in an analysis of the intervention group only, there was a decrease between Time 2 and Time 3. It may be that worries that were school-related decreased because there was a summer vacation between Time 2 and Time 3. Furthermore, Time 1 was prior to regular examinations, and the results of the examinations were imminent at Time 2, so it can be assumed that these were periods of increased worry regarding school-related issues. As such, this may reflect a change in school events rather than a change arising from intervention effects.

The changes at the three time points in the intervention group only showed an increase in the ratio of students who had reliable adults and adults who they can talk to at any time. However, there was no change in terms of the adults who care about them and adults who greet them. These differences reflect the content of the program. “Reliable adults” and “adults who you can talk to at any time” were emphasized in the program, and it included wording that promoted these concepts. However, in the course of the program, while there was some discussion of “the importance of each person”, there were no opportunities to consider “adults who care about you” and “adults who greet you”, and so this may not have been shown as an intervention effect. A similar inference may apply to the fact that no increase in general self-worth was observed.

Many previous studies targeted mental health knowledge and attitudes and gatekeeper programs and provided evidence for their effectiveness [4,18,19]. To recognize the warning signs for suicide is important; on the other hand, for students to know ways of help-seeking by themselves is also important. Unlike previous studies, our program focused on ways of help-seeking for the neighborhood, including older adults. In addition, the neighborhood actually joined this program so that students could experience its warmth. The Suicide Prevention Resource Center suggests that connectedness to individuals, family, community, and social institutions are protective factors [2]. Our implementation and findings can provide a new avenue for suicidal prevention program by creating social capital.

However, there were two limitations to this study. The first is a limitation in terms of the study design and question items. In the present study, one of the reasons why changes were found in the ratios of those with worries was that the intervention itself produced an awareness of self-disclosure about one’s worries in connection to human relationships. However, as there was no investigation of changes to the experience of reaching out to others during the program implementation period; this is only conjecture, and there is no evidence for the relevant mechanisms. In the future, it will be necessary to clarify this by investigating the changes to behavior within the implementation period. With regard to worries about schoolwork, there is a strong possibility that this was affected by changes in school events such as examination periods and summer vacations. In order to more clearly verify the effects of the program, a study design should be devised so that school events are as uniform as possible during the program. The uniformity of events can be facilitated by establishing a program period across half a year or a whole year. This will also enable a group comparison across the three time points, which was not possible in this study.

The second limitation was the content of the program. This program largely did not address worries about schoolwork. Examining students’ worries by type indicates that there were more worries about schoolwork than human relationships. In the sense of responding to the needs of school-aged young people, it will be necessary to make revisions so that the program promotes the resolution of worries that are school-related.

In future studies, the differences of the impact of the program between school grades must also be taken into consideration. While this study was limited to second-year junior high school students, it is also possible that newly-enrolled first-year students struggle to find people who they can reach out to. In the third year, students prepare for high school entrance exams, so they may have more worries about schoolwork, which makes an approach to worries about schoolwork more important still. For that reason, there is a need for longitudinal observation and long-term verification of effects. At the same time, implementing a survey and qualitative study that clarifies the types of adult that young people find easier to talk to can be expected to increase the quality of SOS output education programs in the future.

## 5. Conclusions

In this study, regions (elderly volunteers), governments (municipal staff), and schools (educators) cooperated to implement an SOS output education program to investigate the impact of the program on support-seeking awareness and behavior among junior high school students.

As a result, it was found that there was an impact on the awareness (self-disclosure) among students regarding seeking support in connection to human relationships. Additionally, there was an increase in students who know “reliable adults” and “adults who you can talk to at any time”. However, due to the limitations of the study design and question items, the mechanisms of those changes and the differences between school grades could not be investigated. Furthermore, in some cases, there were no effects observed in connection to the type of worries or the type of person that students were seeking support from. In the future, it will be necessary to expand the scope of the significant effects of the program and to revise the program for that purpose.

## Figures and Tables

**Figure 1 children-09-00541-f001:**
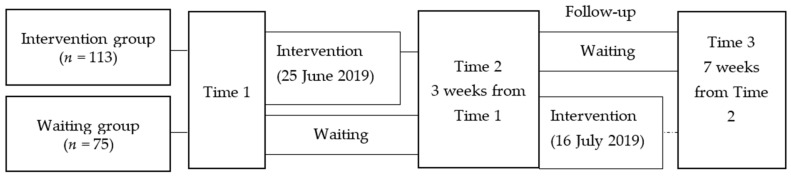
Flow of participants and procedures.

**Table 1 children-09-00541-t001:** Contents of program.

Study Activity/Content
**Part 1 Introduction (5 min)**
**Introduction** Explanation of health center and lesson content.
**Part 2 Stress coping/SOS output lecture (25 min)**
**Aims** Stress awareness; Convey that each person is important. On stress and stress coping: Question 1: “When you have difficult feelings, what do you do in order to lighten your mood? Please write down your coping methods.” · Reveal examples of stress coping (e.g., singing or eating a rice cracker). · “The best thing to do is to talk to a reliable person who you are close to.” How to seek help. Question 2: “When you are struggling, is there someone who you want to talk to about your worries? Give this some thought.” · “It can be difficult to find reliable adults so please try talking to at least three different people.” · “If you are unable to find a reliable adult, you can contact a support center by phone or email.” Question 3: “If your friend was struggling, what would you do to help that friend?” · “Please talk to that friend and find a reliable adult together. Make sure not to blame that person for their situation.” Key message: · Each person is important; · Seek help from a reliable adult when you are suffering.
**Part 3 Picture-book reading (10 min)**
**Picture-book reading** *You Are Special* by Max Lucado
**Part 4 Conclusion (2 min)**
**Distribute support center leaflets.** **Review today’s lesson and write down your thoughts on a piece of paper.**

**Table 2 children-09-00541-t002:** Ratio of “presence of worries” in each group.

	Intervention Group (*n* = 104)			Waiting Group (*n* = 66)		
	Time 1	Time 2	Time 3	Time 1	Time 2	Time 3
Worry (overall)	51.0%	60.6%	47.1%	51.5%	47.0%	50.0%
Studying	35.6%	44.2%	32.7%	33.3%	31.8%	40.9%
Future	22.1%	29.8%	22.1%	30.3%	34.8%	31.8%
Friendships	19.2%	30.8%	17.3%	18.2%	30.3%	16.7%
Romance	5.8%	8.7%	3.8%	6.1%	9.1%	4.5%
Bullying	1.0%	0.0%	0.0%	0.0%	0.0%	0.0%
Sexuality	1.0%	1.9%	1.0%	0.0%	0.0%	0.0%
Family/Relative	5.8%	7.7%	3.8%	6.1%	6.1%	7.6%
Local People/Local Situation	0.0%	1.9%	0.0%	3.0%	3.0%	1.5%
Other	4.8%	2.9%	7.7%	6.1%	3.0%	7.6%
School-Related (Study or Future Path)	39.4%	48.1%	36.5%	39.4%	42.4%	43.9%
Human Relationships (Friendships or Romantic)	20.2%	32.7%	17.3%	21.2%	33.3%	19.7%

**Table 3 children-09-00541-t003:** Ratio of “the presence of familiar adults” in each group.

	Intervention Group (*n* = 104)			Waiting Group (*n* = 66)		
	Time 1	Time 2	Time 3	Time 1	Time 2	Time 3
Reliable adults	62.5%	58.7%	70.2%	66.7%	66.7%	68.2%
Adults who you can talk to at any time	43.3%	49.0%	55.8%	47.0%	54.6%	54.6%
Adults who care about you	56.7%	60.6%	59.6%	71.2%	62.1%	62.1%
Adults who greet you	60.6%	64.4%	63.5%	59.1%	60.6%	56.1%

**Table 4 children-09-00541-t004:** Means and standard errors of general self-worth in each group.

	Intervention Group (*n* = 92)			Waiting Group (*n* = 53)		
	Time 1 *M* (*SE*)	Time 2 *M* (*SE*)	Time 3 *M* (*SE*)	Time 1 *M* (*SE*)	Time 2 *M* (*SE*)	Time 3 *M* (*SE*)
General self-worth	23.05 (0.69)	22.93 (0.74)	22.87 (0.71)	22.77 (0.81)	22.49 (0.83)	22.47 (0.70)

## Data Availability

This dataset used and/or analyzed during the current study is available from the corresponding author upon reasonable request.

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
