# Peer review of "Suicide Prevention Program with Cooperation from Senior Volunteers, Governments, and Schools: A Study of the Intervention Effects of “Educational Lessons Regarding SOS Output” Focusing on Junior High School Students"

_children, 2022, doi:10.3390/children9040541_

Round 1
Reviewer 1 Report
The paper is very interesting and investigates the impact of aprogram on support-seeking awareness and behavior among 99 junior high school students. The metodology appears standardizated and the statistical analysis is rigorous. Authors are advised to shorten the introduction because it is too long, explain the results more clearly, and explain the role of research in reducing the risk of suicide.
Author Response
"Please see the attachment."

Reviewer 2 Report
This paper is an interesting report of a suicide prevention program.
Prevention programs like the one reported are so important and necessary. Especially when they involve different age levels and different subjects in a community. In the same way, prevention from early ages as carried out by this program can be very helpful in the prevention of suicidal behavior.
Two are my questions and that I consider can help improve the paper. First. Put the approval data by the ethics committee. As well as the date of completion of the program. Was it implemented, before the COVID-19 pandemic, after or during?
Second. In the discussion, I believe that the paper should search the literature for similar or already implemented programs at the international level. See what other countries are working in this direction and discuss their results in comparison with other reports.
Finally, congratulations to the authors for this important contribution.
Author Response
"Please see the attachment."
